# Gallic Acid Alkyl Esters: Trypanocidal and Leishmanicidal Activity, and Target Identification via Modeling Studies

**DOI:** 10.3390/molecules27185876

**Published:** 2022-09-10

**Authors:** Dietmar Steverding, Lázaro Gomes do Nascimento, Yunierkis Perez-Castillo, Damião Pergentino de Sousa

**Affiliations:** 1Bob Champion Research and Education Building, Norwich Medical School, University of East Anglia, Norwich NR4 7UQ, UK; 2Laboratory of Pharmaceutical Chemistry, Department of Pharmaceutical Sciences, Federal University of Paraíba, João Pessoa 58051-900, PB, Brazil; 3Bio-Cheminformatics Research Group, Universidad de Las Américas, Quito 170516, Ecuador; 4Facultad de Ingeniería y Ciencias Aplicadas, Área de Ciencias Aplicadas, Universidad de Las Américas, Quito 170516, Ecuador

**Keywords:** gallic acid alkyl ester, natural products, *Trypanosoma brucei*, *Leishmania major*, trypanocidal activity, leishmanicidal activity, molecular docking, molecular dynamics simulations

## Abstract

Eight gallic acid alkyl esters (**1–8**) were synthesized via Fischer esterification and evaluated for their trypanocidal and leishmanicidal activity using bloodstream forms of *Trypanosoma brucei* and promastigotes of *Leishmania major*. The general cytotoxicity of the esters was evaluated with human HL-60 cells. The compounds displayed moderate to good trypanocidal but zero to low leishmanicidal activity. Gallic acid esters with alkyl chains of three or four carbon atoms in linear arrangement (propyl (**4**), butyl (**5**), and isopentyl (**6**)) were found to be the most trypanocidal compounds with 50% growth inhibition values of ~3 μM. On the other hand, HL-60 cells were less susceptible to the compounds, thus, resulting in moderate selectivity indices (ratio of cytotoxic to trypanocidal activity) of >20 for the esters **4**–**6**. Modeling studies combining molecular docking and molecular dynamics simulations suggest that the trypanocidal mechanism of action of gallic acid alkyl esters could be related to the inhibition of the *T. brucei* alternative oxidase. This suggestion is supported by the observation that trypanosomes became immobile within minutes when incubated with the esters in the presence of glycerol as the sole substrate. These results indicate that gallic acid alkyl esters are interesting compounds to be considered for further antitrypanosomal drug development.

## 1. Introduction

Trypanosomatids are protozoan parasites that cause various diseases in humans and animals. Species of the genus *Trypanosoma* are responsible for Chagas disease and sleeping sickness in humans and nagana disease in livestock [1,2], and species of the genus *Leishmania* for different forms of cutaneous and visceral diseases in humans [3]. These parasites are transmitted to their mammalian host by insect vectors, which in the case of African trypanosomes, are tsetse flies, in the case of *T. cruzi*, are kissing bugs, and in the case of *Leishmania* sp., are sandflies. Treatment of trypanosomatid diseases relies solely on chemotherapy, but most licensed drugs are outdated and not very effective [4]. In addition, the development of drug resistance in trypanosomatid parasites is a growing problem, particularly in trypanosomes infecting livestock [5]. For these reasons, the search for new drug candidates with the potential to be developed into effective treatments of trypanosomatid diseases is urgently needed.

Natural products have been the source of numerous approved drugs and have been shown to exhibit potent antiproliferative activity against trypanosomatids [6,7]. Phenolic acids are a promising class of natural products that have previously been found to have antimicrobial activities [8]. A few phenolic acids, caffeic acid, gallic acid, and rosmarinic acid, have also been discovered to display trypanocidal activity [9,10]. Interestingly, esterification of caffeic acid results in compounds with much increased trypanocidal activity [9,11]. Moreover, the introduction of a third hydroxyl group in the aromatic ring seems to increase the inhibitory activity of caffeic acid esters [12]. These previous findings prompted us to investigate the trypanocidal and leishmanicidal activities of alkyl esters of the 3,4,5-trihydroxy phenolic acid, gallic acid (3,4,5-trihydroxybenzoic acid). In addition, modeling studies were carried out to identify potential targets for gallic acid alkyl esters.

## 2. Results and Discussion

### 2.1. Synthesis and Characterization of Gallic Acid Alkyl Esters

Gallic acid alkyl esters **1**–**8** were prepared by acid-catalyzed esterification of gallic acid with alkyl alcohols under solvent-free conditions, i.e., the alcohol serves as solvent and reactant at the same time (Figure 1). All compounds were readily purified by silica gel column chromatography in high yields ranging between 50 and 90%. The compounds were identified based on their melting points, R*_f_*-values obtained from thin-layer chromatography, and IR, ^1^H-NMR, and ^13^C-NMR spectra, by comparison with the literature data [13,14,15]. Spectroscopic data confirmed that the 3,4,5-trihydroxybenzoate substructure was maintained for all ester products. Compared with gallic acid, the IR spectra of the esters showed a slight shift of the C=O stretching band from 1668 cm^−1^ (gallic acid; [16]) to 1671–1707 cm^−1^. This shift was dependent on the alkyl group, with the iso-alkyl groups producing the smallest changes.

### 2.2. Biological Activity of Gallic Acid Alkyl Esters

All eight gallic acid alkyl esters **1–8** inhibited the growth of bloodstream forms of *T. brucei* in a dose-dependent manner, with minimal inhibitory concentration (MIC) values ranging from 10–100 μM and 50% growth inhibition (GI_50_) values ranging from 3–33 μM (Table 1). The most trypanocidal gallic acid alkyl esters were compounds **4**, **5**, and **6**, followed by derivatives **3** and **8**. These five esters were 1.7- to 4.7-fold more trypanocidal than the reactant gallic acid (GI_50_ = 14.2 μM [10]), indicating that esterification of this phenolic acid can generate compounds with improved antitrypanosomal activity. Compared with suramin, one of the drugs used in the treatment of sleeping sickness, the three most trypanocidal compounds **4**, **5**, and **6** were 10–100 times less active (Table 1).

In contrast to bloodstream-form trypanosomes, *L. major* promastigotes were much less sensitive toward the gallic acid alkyl esters (Table 2). Compounds **2** and **7** displayed no leishmanicidal activity, while gallic acid esters **4** and **5** were the only compounds for which a GI_50_ value could be determined. Based on MIC values, the gallic acid alkyl esters were >1000 times less leishmanicidal than the antileishmanial drug amphotericin B (Table 2). However, the overall inhibition trend of the gallic acid alkyl esters was similar between the two parasite species, i.e., compounds with potent trypanocidal activity also displayed higher leishmanicidal activity, while less active trypanocidal compounds exhibited zero to low leishmanicidal activity.

The gallic acid alkyl esters showed low cytotoxic activity against HL-60 cells (Table 1). All compounds had a MIC value of >100 μM and GI_50_ values of >75 μM. Gallic acid esters **1**, **2**, and **7** seemed to display no cytotoxicity against the human cells. Despite the low cytotoxic activity, the gallic acid alkyl esters’ selectivity (ratio of cytotoxic to trypanocidal activity) was only moderate (Table 3). The compounds with the best MIC and GI_50_ ratios of >10 and from 22–28 were gallic acid alkyl esters **4**, **5**, and **6**. In contrast, the antitrypanosomal drug suramin has 10 times higher MIC ratios and 100 times higher GI_50_ ratios (Table 3).

Structure-activity relationship analysis indicates that there is no correlation between the lipophilicity of the different gallic acid alkyl esters and their trypanocidal activity. As shown in Figure 2A, predicted log *P* values as a measure for lipophilicity of the compounds did not correlate with their GI_50_ values. On the other hand, a correlation was found between the water solubility of the different compounds and their antitrypanosomal activity. According to Figure 2B, predicted log *S* values as a measure for water solubility of the esters showed some association with their GI_50_ values. Based on these findings, water solubility appears to be a weak predictor for the trypanocidal activity of gallic acid alkyl esters. Furthermore, it seems that the length of the alkyl group influences the activity of the gallic acid esters. Compounds with an alkyl chain of three or four carbon atoms in linear arrangement (gallic acid propyl (**4**), butyl (**5**), and isopentyl (**6**) ester, respectively) were the most trypanocidal agents. Additionally, compound **8** with a 2-methoxyethyl chain containing three carbon atoms and one oxygen atom in a linear arrangement is in accordance with this rule, although its antitrypanosomal activity is slightly lower than that of the potent compounds **4**–**6**. On the other hand, gallic acid esters with shorter (one or two carbon atoms; gallic acid methyl (**1**) and ethyl (**2**) ester) or longer (five carbon atoms; gallic acid pentyl ester (**7**)) alkyl chains were approximately ten times less trypanocidal. Only compound **3** seems not to fit with this pattern as it has an alkyl chain with two carbon atoms in a linear arrangement (isopropyl) but exhibited three times greater antitrypanosomal activity than ethyl gallic acid. This structure-activity relationship confirms previous findings obtained with two related phenolic esters, 3,4-dihydroxycinnamic (caffeic) acid isopentyl ester and 3-methoxy-4-hydroxycinnamic (ferulic) acid ethyl ester [9]. Whereas caffeic acid isopentyl ester was shown to display potent trypanocidal activity with a GI_50_ value of 1.24 μM, ferulic acid ethyl ester was found to exhibit much lower antitrypanosomal activity with a GI_50_ value of 110 μM [9]. However, the structure-activity pattern found in this study for the trypanocidal action of gallic acid esters differs from that previously determined for the antibacterial action of alkyl gallates. Potent bactericidal activity was observed for gallic acid esters with longer alkyl chains of between eight and twelve carbon atoms [17,18,19,20]. Similarly, the antifungal activity of gallic acid esters was associated with the C6 to C9 alkyl chain [21]. On the other hand, gallic acid esters with longer alkyl chains seem to be more cytotoxic than those with shorter alkyl chains. For example, octyl (C10) and dodecyl (C12) gallates display potent cytotoxic activity against murine B-lymphoma WEHI-231 cells with GI_50_ values of 1.5 μM and 1.0 μM, respectively [22]. Thus, the trypanocidal activity of gallic acid esters with shorter alkyl chains (C3 and C4) proved to be advantageous as these esters are less cytotoxic.

### 2.3. Target Identification via Molecular Modeling Studies

Modeling studies combining computational target fishing, molecular docking, and Molecular Dynamics (MD) simulations were performed with the objective of identifying potential targets of compound **4** in *T. brucei*. First, the potential targets of the compound were identified through computational target fishing. Then, compound **4** was docked into the identified predicted target proteins. Finally, the top three scored ligand conformations per target were subject to MD simulations, and the free energy of binding was estimated with the Molecular Mechanics Poisson–Boltzmann Surface Area (MM-PBSA) method. These MD-based free energy of binding values were used as the criterion for selecting the most likely targets of compound **4** in *T. brucei*. The objective of the MD simulations was to obtain an ensemble of conformations to be used in MM-PBSA calculations. That is, MD simulations were employed to estimate the energetic stability of the predicted complexes.

To determine potential targets for gallic acid alkyl esters, the Similarity Ensemble Approach (SEA) was employed [25]. Homology-based target fishing [26] was then carried out with the most trypanocidal compound **4**. This fishing approach identified four enzymes, glucose-6-phosphate dehydrogenase (G6PD), protein kinase A catalytic subunit isoform 1 (PKA1), farnesyltransferase (FT), and isoleucine-tRNA ligase (IleRL), as potential targets of compound **4** in *T. brucei*. In addition, the trypanosome alternative oxidase (TAO) was included in the molecular modeling studies because gallic acid alkyl esters share some structural similarities with the classical TAO inhibitors salicylhydroxamic acid (SHAM) and ascofuranone, and, in particular, with various derivatives of ascofuranone (4-alkoxybenzoic acids) [27,28]. Specifically, the Tanimoto coefficient calculated with ChemMine [29] ranged from 0.4000 to 0.5263 for gallic acid alkyl esters and SAHM and ACB41 as representative of 4-alkoxybenzoic acids [27], respectively, indicating that there is a medium similarity between the molecules (0.4–0.7, [30]). In the case of gallic acid alkyl esters and ascofuranone, the Tanimoto coefficient range from 0.2432 to 0.2647 suggesting a low similarity between these compounds (0.2–0.4 [30]). For molecular docking calculations, the cofactor and substrate binding sites were explored separately for G6PD. Likewise, two scenarios were considered for modeling TAO. The first included a hydroxide anion within the enzyme structure, while the second considered the possibility that the anion is displaced by a ligand molecule and, hence, was removed from the enzyme prior to modeling. These two scenarios are possible for TAO and have been supported by experimental X-ray structures of the enzyme bound to inhibitors [31].

Molecular docking studies were carried out as described in the Section 3. Before applying the docking protocol to compound **4**, it was tested whether it could reproduce the experimental binding modes of two inhibitors determined from co-crystallized complexes with TAO. These crystal structures are complexes of TAO with colletochlorin B (PDB code 3W54 [32]) and the coumarin derivative 7,8-dihydroxy-4-[[4-(4-methoxyphenyl)piperazin-1-yl]methyl]chromen-2-one (PDB code 5GN7 [32]). Only these two structures were evaluated since no complexes of any of the other proteins with inhibitors are deposited in the PDB database [32]. These validations were performed starting from the 2D representation of the ligands, following the same protocol described for compound **4**. In both cases, it was possible to obtain docking conformations of the ligands with root mean square deviation (RMSD) values lower than 2 Å, relative to the experimental orientations of the compounds, among the top three scored solutions. This result supports the selected docking methodology and its further application to compound **4**.

The docking scores obtained for the top three ligand conformations per target are presented in Table 4. The highest GOLDScores and CHEMScores were obtained for TAO, indicating a higher binding affinity of compound **4** to this enzyme compared to the other proteins. The three ligand conformations selected for MD simulations on each target as well as the observed interaction networks are given as Appendix A. These results show that, as expected from the implemented molecular docking methodology, there is diversity in the subset of ligand conformations selected for MD simulation in all targets.

Docking scoring functions are designed for the virtual screening of databases of compounds against a single target. Thus, their use to select the potential target of a single compound can lead to biased results [33,34]. This limitation is associated with the simplifications introduced in scoring functions that are required to obtain acceptable accuracy/speed tradeoffs during virtual screening. For this reason, molecular docking was only employed to obtain initial binding hypotheses of compound **4** to its potential targets, but not for the selection of the most likely compound’s targets. For target selection, we used the more accurate free energy of binding obtained with the MM-PBSA method. The refinement of docking solutions with MM-PBSA calculations conducted from MD simulations has proven to produce more reliable estimations of ligand-receptor affinities than docking alone [35,36].

One aspect to consider when MD simulations are used to obtain conformational ensembles for MM-PBSA calculations is the length of the simulations. This is a topic highly discussed in the scientific literature, and there is no consensus on the optimal length of MD simulations for MM-PBSA calculations. Nevertheless, many authors agree that short (less than 5 ns) simulations would be sufficient for MM-PBSA calculations [35,36]. Based on the available evidence, we performed five different 4 ns MD replicas for each of the 21 docking-predicted complexes. With this setup, 20 ns of MD simulations were performed per complex and a total simulation time of 420 ns was achieved across all systems. The five different MD replicas, each one starting with different random initial velocities, ensure a better exploration of the complex’s conformational space compared to a single trajectory approach.

All docking-predicted complexes were subject to MD simulations and the free energy of binding was estimated following the procedure described in the Section 3. The results of the MD-based MM-PBSA calculations are summarized in Table 4. It is interesting to note that the GOLDScore and CHEMScore values reported in Table 4 show a Kendall’s correlation coefficient of 0.56. This is an indication that the rankings produced by both scoring functions are positively correlated. Likewise, Kendall’s correlation between the scoring functions and the MM-PBSA energies are −0.37 and −0.48 for the GOLDScore and CHEMScore, respectively. These negative correlations can be interpreted as positive correlations between the rankings since higher docking scores indicate better binding, and lower MM-PBSA energies suggest higher ligand affinity. Although correlation exists between the rankings produced by the scoring functions and the MM-PBSA energies, important differences can be observed between them. For example, the most energetically stable complex predicted by the MM-PBSA method ranks in 15th and 4th positions according to the GOLDScore and CHEMScore scoring functions, respectively. On the other hand, the complexes ranked in the first three positions according to the CHEMScore function, occupy positions 5, 14, and 19 according to the MM-PBSA energies, respectively. These observations suggest that docking scores should not be used as a target selection criterion in replacement of a more accurate methodology such as MM-PBSA.

The MD simulations showed the lowest free energy for the binding of compound **4** to TAO when the hydroxide anion was present in the enzyme’s active site. Thus, the modeling results suggest that the most probable target of compound **4** in the bloodstream forms of *T. brucei* is TAO. Although TAO was the receptor with the best docking scoring values, it must be considered that the docking protocol ranked the complex without a hydroxide anion first. The predicted binding mode of compound **4** to TAO as well as the observed ligand-enzyme interactions are presented in Figure 3. The structure shown corresponds to the centroid of the most populated cluster obtained after grouping 100 MD snapshots used for the MM-PBSA calculations. The predicted binding pose of compound **4** to TAO shows a large network of interactions between compound **4** and the enzyme. The 3,4,5-trihydroxybenzoate substructure orientates toward the bottom of the enzyme’s active site cavity, forming hydrogen bonds with the hydroxide anion and the side chain Y220. This moiety is flanked by several hydrophobic residues such as A216, C119, L122, A126, and T219. In addition, the central carbonyl oxygen of the compound is predicted to form a hydrogen bond with the side chain R118. The predicted hydrogen bonds are proposed as the main factors stabilizing the compound-enzyme complex. Finally, the alkyl tail points to the helix 83–103, limiting the active site’s size and accommodating a small cavity lined by V92, C95, R96, F99, and L212.

The orientation of the alkyl group in the binding cavity could explain the observed structure-activity relationship observed for the gallic acid alkyl esters. According to the structural model, the cavity can optimally accommodate linear chains of length between 3 and 4 carbon atoms. Longer chains could lead to steric hindrance within the binding cavity, while shorter chains may have reduced contact with residues in the enzyme’s active site. In both cases, the energetic stability of the complexes would be reduced either due to reduced compound-enzyme interactions or due to steric constraints. In the specific case of compound **3**, its branched chain allows for more contact with the enzyme compared to compound **2** with an ethyl chain. This could explain the improved trypanocidal activity of compound **3** over compound **2**.

To further assess the proposed inhibition of TAO by compound **4**, the same MD simulation and MM-PBSA calculations were applied to estimate the free binding energy of the potent TAO inhibitors, colletochlorin B and 7,8-dihydroxy-4-[[4-(4-methoxyphenyl)piperazin-1-yl]methyl]chromen-2-one [31,39]. The predicted free binding energy of the inhibitors in the complex with TAO was calculated to be −11.61 kcal/mol and −12.31 kcal/mol, respectively, which is similar to that estimated for the compound **4**-TAO complex. This finding further supports the suggestion that gallic acid alkyl esters are inhibitors of TAO.

### 2.4. ADMET and Druglikeness Properties of n-Propyl Gallate (Compound ***4***)

Computational predictions were also performed for the ADMET properties of compound **4** and the reference trypanocidal drug suramin. These predictions are listed in Table 5 and were obtained with the SwissADME [24] and pkCSM web servers [40]. SwissADME was employed to predict the physicochemical properties and for the PAINS analysis, while the rest of the reported predictions were obtained with the pkCSM server. The first observation from these analyses is that compound **4** is predicted as PAINS due to the presence of a catechol substructure [41]. As recommended in the scientific literature, before proceeding to any future optimization of compound **4** as a trypanocidal agent, it is necessary to fully clarify if it is indeed a PAINS [42]. In addition, future hit-to-lead optimization campaigns must lead to compounds where such PAINS alerts are eliminated.

In contrast to suramin, compound **4** has suitable physicochemical parameters for oral bioavailability. Another advantage of compound **4** over the reference chemical, is that it is predicted to have high gastrointestinal absorption. Both compounds are proposed to be skin permeable, poorly distributed to the brain, and unable to penetrate the central nervous system. Likewise, neither compound seems to be a cytochrome P450 inhibitor or substrate. In terms of toxicity, both compounds show a similar profile, despite the predicted tolerated dose of compound **4** being low. Given that compound **4** is a hit chemical, the ADMET property predictions should be considered in the future optimization of its trypanocidal activity.

According to SwissADME, compound **4**, like all the other gallic acid alkyl esters, is predicted to be a drug-like molecule. The bioavailability score of compounds **1–8** is estimated with SwissADME to be 0.55.

### 2.5. Effect of Gallic Acid Alkyl Esters on the Motility of Trypanosomes

Proliferating bloodstream forms of *T. brucei* rely exclusively on glycolysis for energy production [43]. Recently it has been shown that glycerol can also support the growth of *T. brucei* bloodstream forms [44]. However, with glycerol as the sole substrate, inhibition of TAO leads immediately to immobility of the bloodstream-form trypanosomes [45]. With glucose as substrate, however, inactivation of TAO does not affect the motility of bloodstream-form trypanosomes [45]. The reason for this is that with glycerol as a substrate, inhibition of TAO leads to blockage of ATP synthesis while in the presence of glucose, the ATP level remains about half of that found in the absence of TAO inhibition [45]. The incubation of bloodstream forms of *T. brucei* with gallic acid alkyl esters **1**–**8** in the presence of glycerol also led to immobility of the cells within 5 min (Table 6). Importantly, when no inhibitor was present, the cells remained motile with glycerol as the substrate (Table 6). Additionally, in the presence of glucose, the motility of bloodstream-form trypanosomes was not impaired by the compounds (Table 6). This observation supports the finding of the MD studies that TAO is most likely the target of gallic acid alkyl esters.

### 2.6. Conclusions

This study has shown that the esterification of gallic acid can yield compounds with improved trypanocidal activity. With GI_50_ values of ~3 μM (0.63–0.78 μg/mL) and selectivity indices (GI_50_ ratios) of >20, the gallic acid alkyl esters **4**, **5**, and **6** are not far off from meeting the activity and cytotoxicity criteria for drug candidates for African trypanosomiasis (GI_50_ < 0.2 μg/mL; selectivity > 100 [46]). Regarding the selectivity, it should be pointed out that the HL-60 cells used in this study in determining the cytotoxic action of the compounds are cancer cells, and, therefore, the cytotoxicity of the gallic acid alkyl esters has likely been overestimated. For instance, compounds **4** and **5** have previously been shown to exhibit much lower cytotoxicity against Vero cells [47], which are non-cancerous cells. Compared with HL-60 cells, Vero cells are 8.7 and 4.8 times less sensitive to gallic acid propyl ester (GI_50_(Vero) = 713 μM) and gallic acid butyl ester (GI_50_(Vero) = 361 μM), respectively [47]. Thus, when using the Vero cell cytotoxicity as the basis, compounds **4** and **5** will meet the selectivity criteria of >100.

Much evidence indicates that TAO is the target of gallic acyl alkyl esters. First, molecular modeling studies revealed that the compound **4**-TAO complex has the lowest free binding energy. Second, with glycerol as the substrate, the motility of bloodstream-form trypanosomes is blocked by gallic acyl alkyl esters. Third, gallic acyl alkyl esters display very low leishmanicidal activity against promastigotes of *L. major*. Unlike proliferating bloodstream forms of *T. brucei*, promastigotes of *L. major* do express an electron transport chain and, therefore, should not be affected by the inhibition of TAO. The ultimate proof that TAO is the target of gallic acid alkyl esters would be the demonstration that the respiration in bloodstream-form trypanosomes is inhibited by the compounds.

The modeling results may help in designing gallic acid esters with better binding activity against TAO and improved trypanocidal activity. For example, the predicted binding mode suggests that the introduction of a substituent capable of forming a hydrogen bond at the methyl group of the alkyl chain of compound **4** could increase the stability of the compound-TAO complex. Furthermore, X-ray crystal structure analysis of TAO bound to the coumarin derivative 7,8-dihydroxy-4-[[4-(4-methoxyphenyl)piperazin-1-yl]methyl]chromen-2-one indicates that the enzyme may be able to accommodate gallic acid with more bulky substituents, e.g., aryl groups. This suggestion is supported by the potent trypanocidal activity of caffeic acid phenethyl ester displaying a GI_50_ value of 0.046 μM [11]. Whether gallic acid aryl esters would have improved trypanocidal activity remains to be shown.

One limitation of the proposed modeling approach is that the ligand conformational entropy is neglected in the calculation of the MM-PBSA energies. This computation is usually performed by normal-mode analysis as it is a highly computationally intensive task and is often omitted during MM-PBSA calculations [48]. Despite ignoring the entropic term during the modeling process, we consider that the modeling results are valuable since they provide a binding hypothesis of compound **4** to TAO that is consistent with the obtained experimental results. The proposed model could be the starting point for future computer-guided optimization of gallic acid aryl esters as trypanocidal agents using more accurate modeling approaches.

## 3. Materials and Methods

### 3.1. Chemistry

All reagents were purchased from Sigma Aldrich (St. Louis, MI, U.S.) and were of commercial grade. IR spectra were recorded on an FTIR Cary 630 (Agilent Technologies, Santa Clara, CA, USA) spectrometer. ^1^H and ^13^C-NMR spectra were recorded either on a Varian Mercury spectrometer at 200 MHz and 50 MHz, respectively, or a Bruker BioSpin spectrometer at 400 MHz and 100 MHz, respectively. Chemical shifts were reported relative to the DMSO-d_6_ solvent peak.

The general procedure for the synthesis of gallic acid alkyl esters was as follows: To a mixture of gallic acid (0.1 g, 0.59 mmol) in 10 mL of alkyl alcohol to be esterified, 0.5 mL of concentrated H_2_SO_4_ was added. The solution was stirred under reflux for 3 to 7 h, and the progress of esterification was monitored by thin-layer chromatography. Once the reaction was completed, excess alcohol was evaporated under reduced pressure, and the crude product was diluted into 10 mL ethyl acetate and washed with 15 mL water. After separating the organic phase, the aqueous phase was extracted three times with 10 mL ethyl acetate, and the combined organic phases were treated with 10 mL aqueous 5% NaHCO_3_ solution. The organic phase was dried with anhydrous Na_2_SO_4_, filtered, and evaporated under reduced pressure. The pure product was obtained by silica gel column chromatography (eluent: hexane/ethyl acetate 1:1) [49].

Gallic acid methyl ester (**1**): White solid (98 mg; 0.53 mmol), 90.52% yield; MP = 199–200 °C (lit. 200–202 °C [13]); TLC (1:1 hexane/EtOAc), R*_f_* = 0.64. ^1^H-NMR (400 MHz, DMSO-d_6_), δ_H_ 6.93 (s, 2H, H-2, H-6), 3.74 (s, 3H). ^13^C-NMR (100 MHz, DMSO-d_6_) δ_C_ 166.5, 145.7, 138.6, 119.4, 108.6, 51.7. IR v_max_ (KBr, cm^−1^) 3368, 3220, 3019, 1692, 1618, 1444, 1373 [13].

Gallic acid ethyl ester (**2**): Brown solid (107 mg; 0.54 mmol), 91.85% yield; MP = 149–150 °C (lit. 148–150 °C [13]); TLC (1:1 hexane/EtOAc), R*_f_* = 0.66. ^1^H-NMR (200 MHz, DMSO-d_6_) δ_H_ 9.30 (s, 2H, 3,5-OH), 8.96 (s, 1H, 4-OH), 6.95 (s, 2H, H-2, H-6), 4.19 (quart, J = 6.80 Hz, 2H, H-1′), 1.26 (t, J = 6.80 Hz, 3H, H-2′). ^13^C-NMR (50 MHz, DMSO-d_6_) δ_C_ 166.0, 145.7, 138.5, 119.7, 108.6, 60.2, 14.4. IR v_max_ (KBr, cm^−1^) 3292, 3230, 2975, 1707, 1620, 1319 [13].

Gallic acid isopropyl ester (**3**): White solid (97.6 mg; 0.46 mmol), 76.71% yield; MP = 145–146 °C; TLC (1:1 hexane/EtOAc), R*_f_* = 0.87. ^1^H-NMR (400 MHz, DMSO-d_6_), δ_H_ 9.25 (s, 2H, 3,5-OH), 8.89 (s, 1H, 4-OH), 6.94 (s, 2H, H-2, H-6), 5.02 (hept, J = 6.40 Hz, 1H, H-1′), 1.25 (d, J = 6.40 Hz, 6H, H-2′, H-3′). ^13^C-NMR (100 MHz, DMSO-d_6_), δ_C_ 165.4, 145.6, 138.4, 120.1, 108.6, 67.3, 21.9. IR v_max_ (KBr, cm^−1^) 3336, 3109, 2963, 1690, 1616, 1467, 1310 [14].

Gallic acid propyl ester (**4**): White solid (112 mg; 0.53 mmol), 89.79% yield; MP = 145–147 °C (lit. 145–146 °C [13]); TLC (1:1 hexane/EtOAc), R*_f_* = 0.87. ^1^H-NMR (400 MHz, DMSO-d_6_) δ_H_ 9.26 (s, 2H, 3,5-OH), 8.92 (s, 1H, 4-OH), 6.96 (s, 2H, H-2, H-6), 4.11 (t, J = 6.40 Hz, 2H, H-1′), 1.66 (sext, J = 7.60 Hz, 2H, H-2′), 0.94 (t, J = 7.60 Hz 3H, H-3′). ^13^C-NMR (100 MHz, DMSO-d_6_) δ_C_ 166.0, 145.7, 138.5, 119.7, 108.5, 65.6, 21.8, 10,5. IR v_max_ (KBr, cm^−1^) 3291, 3230, 2965, 1707, 1622, 1320 [13].

Gallic acid butyl ester (**5**): White solid (109 mg; 0.48 mmol), 81.97% yield; MP = 125–127 °C (lit. 126–127 °C [13]); TLC (6:4 hexane/EtOAc), R*_f_* = 0.86. ^1^H-NMR (400 MHz, DMSO-d_6_) δ_H_ 6.95 (s, 2H, H-2, H-6), 4.16 (t, J = 6.40 Hz, 2H, H-1′), 1.63 (pent, J = 7.20 Hz, 2H, H-2′), 1.40 (sext, 7.20 Hz, 2H, H-3′), 0.91 (t, J = 7.20 Hz, 3H, H-4′). ^13^C-NMR (100 MHz, DMSO-d_6_) δ_C_ 165.9, 145.6, 138.4, 119.7, 108.5, 63.7, 30.4, 18.9, 13.7. IR v_max_ (KBr, cm^−1^) 3361, 3109, 2958, 1695, 1611, 1341 [13].

Gallic acid isopentyl ester (**6**): White solid (112 mg; 0.47 mmol), 77.74% yield; MP = 110–112 °C; TLC (6:4 hexane/EtOAc), R*_f_* = 0.84. ^1^H-NMR (400 MHz, DMSO-d_6_) δ_H_ 6.94 (s, 2H, H-2, H-6), 4.19 (t, J = 6.80 Hz, 2H, H-1′), 1.76–1.65 (m, 1H, H-3′), 1.55 (quart, J = 6.80 Hz, 2H, H-2′) 0.91 (d, J = 6.40 Hz, 6H, H-4′, H-5′). ^13^C-NMR (100 MHz, DMSO-d_6_) δ_C_ 165.9, 145.6, 138.4, 119.6, 108.5, 62.5, 37.1, 24.8, 22.4. IR v_max_ (KBr, cm^−1^) 3468, 3329, 2960, 1671, 1614, 1339 [14].

Gallic acid pentyl ester (**7**): Light brown solid (108 mg; 0.45 mmol), 76.47% yield; MP = 94–96 °C (lit. 93–94 °C [13]); TLC (6:4 hexane/EtOAc), R*_f_* = 0.81. ^1^H-NMR (400 MHz, DMSO-d_6_); δ_H_ 6.95 (s, 2H, H-2, H-6), 4.14 (t, J = 6.40 Hz, 2H, H-1′), 1.65 (m, 2H, H-2′), 1.33 (m, 4H, H-3′, H-4′), 0.88 (t, J = 6.40 Hz, 3H, H-5′). ^13^C-NMR (100 MHz, DMSO-d_6_) δ_C_ 165.9, 145.6, 138.4, 119.7, 108.5, 64.1, 28.1, 27.8, 21.9, 14.0. IR v_max_ (KBr, cm^−1^) 3362, 3209, 3110, 1695, 1611, 1328 [13].

Gallic acid 2-methoxyethyl ester (**8**): Yellow solid (67.4 mg; 0.30 mmol), 50.24% yield; MP = 152–153 °C (lit. 152–154 °C [15]); TLC (3:7 hexane/EtOAc), R*_f_* = 0.88. ^1^H-NMR (400 MHz, DMSO-d_6_) δ_H_ 6.95 (s, 2H, H-2, H-6), 4.28 (t, J = 4.80 Hz, 2H, H-1′), 3.60 (t, J = 4.80 Hz, 2H, H-2′), 3.29 (s, 3H, H-3′). ^13^C-NMR (100 MHz, DMSO-d_6_) δ_C_ 165.9, 145.6, 138.5, 119.3, 108.6, 70.0, 63.3, 58.2. IR v_max_ (KBr, cm^−1^) 3328, 3036, 2930, 1699, 1627, 1312 [15].

### 3.2. In Vitro Toxicity Assays

Trypanocidal, leishmanicidal, and cytotoxic activities of gallic acid alkyl esters were determined with bloodstream forms of *T. brucei* (clone 427-221a [50]), promastigotes of *L. major* (strain MHOM/IL/81/Friedlin [51]), and human myeloid HL-60 cell [52], respectively. The viability of cells was evaluated with the vital dye resazurin as previously described with some modifications [53,54]. Cells were seeded in 96-well plates in a final volume of 200 μL Baltz medium (*T. brucei* bloodstream forms and HL-60 cells) or Schneider’s insect medium (*L. major* promastigotes) supplemented with 16.7% and 10% fetal bovine serum, respectively. Test compounds were assayed at tenfold dilutions starting from 100 μM down up to 100 nM in the presence of 0.9% DMSO. Wells containing medium with 0.9% DMSO alone served as controls. The initial cell densities were 1 × 10^4^/mL for *T. brucei* bloodstream forms, 2.5 × 10^5^/mL for *L. major* promastigotes, and 5 × 10^4^/mL for HL-60 cells. The cultures were incubated at 37 °C (*T. brucei* bloodstream forms and HL-60 cells) and 27 °C (*L. major* promastigotes) in a humidified atmosphere containing 5% CO_2_. After 24 h of incubation, 20 µL of a 0.5 mM resazurin solution prepared in sterile PBS was added to each well, and the cultures were incubated for another 48 h. Then, the absorbance of each well was read on a BioTek ELx808 microplate reader using a test wavelength of 570 nm and a reference wavelength of 630 nm. The 50% growth inhibition (GI_50_) value, i.e., the concentration of a compound necessary to reduce the growth rate of cells by 50% compared to the control, was determined by linear interpolation [55]. The minimum inhibitory concentration (MIC) value, i.e., the concentration of a compound at which all cells were killed, was determined microscopically.

### 3.3. Motility Assay

A culture of bloodstream forms of *T. brucei* was divided into two equal portions (9 mL) and collected by centrifugation. The cell pellets were resuspended in 1 mL PBS containing 55 mM glucose or 55 mM glycerol. After subsequent centrifugation, the cell pellets were resuspended again in PBS/55 mM glucose and PBS/55 mM glycerol, respectively, and the cell density was adjusted to 2 × 10^6^/mL. Then, 100 μL of trypanosomes were mixed with 100 μL PBS/55 mM glucose or 100 μL PBS/55 mM glycerol containing gallic acid alkyl esters at a concentration of 400 μM, giving a final concentration of the esters in the assay of 200 μM. The final concentration of DMSO in each test was 0.9%. The motility of the trypanosomes was examined under the microscope.

### 3.4. Modeling Studies

Potential targets of the most potent compound **4** were selected following the homology-based target fishing approach previously employed [26]. In brief, targets for the compound were identified with the Similarity Ensemble Approach (SEA) [25]. Then, a BLAST search was performed to find homologous proteins of the SEA predicted targets in *T. brucei*. Any protein in *T. brucei* with a minimum identity of 40% to the SEA predicted proteins, and with at least 75% of its sequence covered by the BLAST alignment, was selected for the modeling studies. In addition, TAO was included in the modeling studies because gallic acid alkyl esters are structurally related to previously reported inhibitors of the enzyme [28]. Ideally, the identification of homologous proteins in *T. brucei* should be performed by considering only residues of the proteins’ binding sites. However, to the best of our knowledge, no method is available to automatically screen whole proteomes and find homologous proteins based on the identity of the binding sites alone.

Among the selected enzymes, only TAO had three-dimensional structures deposited in the RCSB Protein Data Bank (PDB). For this enzyme, the structure deposited in the PDB under the code 3W54 was selected for the modeling studies [32]. The structural models for the other enzymes were obtained from the SWISS-MODEL web server [56]. Different models were generated for each target sequence and the one with the highest QMEANDisCo global score was selected for the modeling studies.

Any modeling parameter not described below in this section was set to the software’s default values. An initial 3D conformation was generated for compound **4** and all hydrogen atoms were added to the compound with the OMEGA algorithm using OpenEye Scientific software [57,58]. Partial atomic charges of the type “am1bcc” were added to the 3D conformer with Molcharge [58].

Molecular docking calculations were performed with the GOLD software [59] using its Hermes interface. Hydrogen atoms were added to the receptor. Only functional relevant ions and cofactors were kept in the receptor. The ligand binding cavity was defined from the co-crystallized ligands for TAO and from the ligands present in the homology models’ templates. A total of 30 different docking solutions were generated for each potential molecular target with the search efficiency parameter set to 200%. The GOLDScore scoring function was selected for primary scoring and rescoring of each predicted inhibitor pose was carried out with the CHEMScore function. The GENERATE diverse solutions option of GOLD was activated while the ALLOW early termination option was disabled. Docking solutions were clustered at an RMSD cutoff of 2 Å. The top three scoring solutions per target according to CHEMScore belonging to different clusters were further analyzed. The post-processing of these three ligand binding poses consisted of MD simulations and the estimation of the free energy of binding from a conformational ensemble extracted from these simulations.

MD simulations were performed with Amber 22 [60] following the procedure previously described [61]. The ff19SB and gaff2 force fields were employed to parametrize proteins and compound **4**, respectively. Parameters for cofactors were obtained from the Amber parameter database [62]. For the TAO metalloenzyme, parameters for the di-iron coordinating region were derived with the Metal Centre Parameter Builder (MCPB) utility of Amber 22 [63]. Parametrized complexes were enclosed in truncated octahedron boxes that were solvated with OPC water molecules. Excess charges were neutralized by the addition of sodium and chloride counterions at an ionic strength of 150 mM according to the previously described methodology [64]. Next, the complexes were energy minimized in two stages, with all atoms except the solvent constrained during the first of these. Energy minimization took place at constant volume and with long-range electrostatic interactions treated with the Particle Mesh Ewald (PME) method. The first energy minimization stage consisted of 500 steps of the steepest descent method followed by 500 cycles of a conjugate gradient. For the second energy minimization, all constraints were released andf 500 steps of the steepest descent algorithm followed by 1000 cycles of a conjugate gradient were conducted. The energy minimized systems were heated for 20 ps from 0 K to 300 K, keeping the solute constrained with a force constant of 10 kcal/mol·Å^2^. From this step on, the bonds involving hydrogen atoms were constrained with the SHAKE algorithm, and the temperature was controlled by a Langevin thermostat with a collision frequency of 1.0 ps^−1^. The final step of systems preparation consisted of equilibration for 100 ps in the NTP ensemble with pressure set to 1 bar and temperature set to 300 K. The equilibrated systems were used as input for the production runs. Five different production runs lasting for 4 ns each were run for each complex. The final free energy of binding wws estimated over 100 MD snapshots evenly extracted from the five MD production runs with the MM-PBSA method as implemented in Amber 22. The internal dielectric factor and ionic strength were set to 2 and 150 mM, respectively, for MM-PBSA calculations.

## Figures and Tables

**Figure 1 molecules-27-05876-f001:**
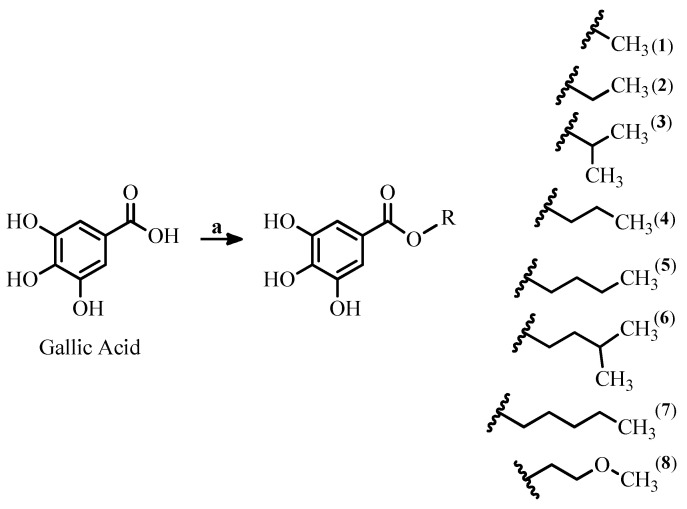
Procedure for preparing gallic acid alkyl esters. Reagents and conditions: (a) ROH, H_2_SO_4_ (cat.), reflux, 3–7 h.

**Figure 2 molecules-27-05876-f002:**
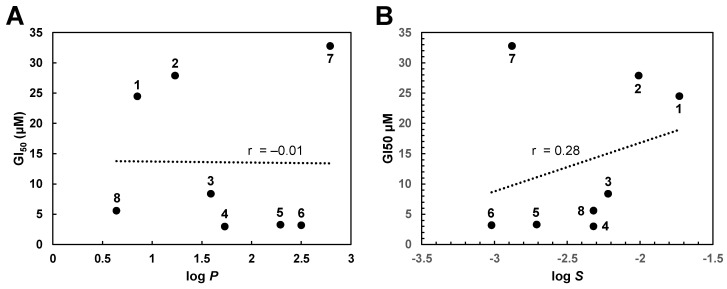
Correlation between predicted log *P* (**A**) and log *S* (**B**) values and GI_50_ values of gallic acid alkyl esters. The numbers shown refer to the individual gallic acid alkyl esters. Predicted log *P* values of gallic acid alkyl esters **1–8** were calculated using the *Interactive logP Calculator* from Molinspiration Cheminformatics [23]. As the correlation coefficient is −0.01 (−0.1–0.0), there is no association between the log *P* values and the GI_50_ values. Predicted log *S* values of gallic acid alkyl esters **1–8** were determined with SwissADME [24]. With a correlation coefficient of 0.28 (0.1–0.39), there is a weak association between the log *S* values and the GI_50_ values.

**Figure 3 molecules-27-05876-f003:**
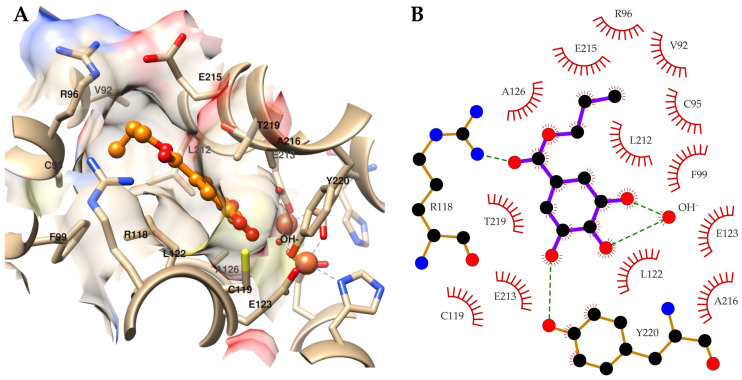
Predicted binding mode (**A**) and network of intermolecular interactions (**B**) of the compound **4**-TAO complex. (**A**) Compound **4** is colored orange and represented as balls and sticks. Only amino acids of TAO interacting with the compound are labeled. (**B**) Only the atoms of compound **4** and residues of TAO forming hydrogen bonds with the compound are shown. The figure was produced with UCSF Chimera and LigPlot+ [37,38].

**Table 1 molecules-27-05876-t001:** In vitro trypanocidal activity of gallic acid alkyl esters against bloodstream forms of *T. brucei* and human myeloid HL-60 cells.

Compound	Alkyl Chain	*T. brucei*		HL-60	
		MIC (μM)	GI_50_ (μM)	MIC (μM)	GI_50_ (μM)
**1**	methyl	100	24.5 ± 4.5	>100	>100
**2**	ethyl	100	27.9 ± 2.4	>100	>100
**3**	isopropyl	100	8.4 ± 2.2	>100	87.9 ± 8.2
**4**	*n*-propyl	10	3.0 ± 0.1	>100	82.0 ± 8.2
**5**	*n*-butyl	10	3.3 ± 0.1	>100	74.8 ± 4.5
**6**	isopentyl	10	3.2 ± 0.1	>100	90.4 ± 21.6
**7**	*n*-pentyl	100	32.8 ± 0.9	>100	>100
**8**	2-methoxylethyl	100	5.6 ± 1.3	>100	99.2 ± 9.4
Suramin	–	1	0.04 ± 0.0	>100	>100

**Table 2 molecules-27-05876-t002:** In vitro leishmanicidal activity of gallic acid alkyl esters against promastigotes of *L. major*.

Compound	Alkyl Chain	MIC (μM)	Growth Inhibition (% at 100 μM)
**1**	methyl	>100	2
**2**	ethyl	>100	0
**3**	isopropyl	>100	34
**4**	*n*-propyl	100	61 (50.4 μM) ^1^
**5**	*n*-butyl	100	58 (62.4 μM) ^1^
**6**	isopentyl	100	45
**7**	*n*-pentyl	>100	0
**8**	2-methoxylethyl	100	47
Amphotericin B	–	0.1	100 ^2^ (0.04 μM) ^1^

^1^ Values in brackets refer to GI_50_ values. ^2^ Percentage growth inhibition at 10 μM.

**Table 3 molecules-27-05876-t003:** MIC and GI_50_ ratios of cytotoxicity to trypanocidal activity.

Compound	Alkyl Chain	MIC Ratio ^1^	GI_50_ Ratio ^1^
**1**	methyl	>1	>4.1
**2**	ethyl	>1	>3.6
**3**	isopropyl	>1	10.5
**4**	*n*-propyl	>10	27.3
**5**	*n*-butyl	>10	22.7
**6**	isopentyl	>10	28.3
**7**	*n*-pentyl	>1	>3.0
**8**	2-methoxylethyl	>1	17.7
Suramin	–	>100	>2500

^1^ MIC ratio, MIC_(HL-60)_/MIC_(*T. brucei*)_; GI_50_ ratio, GI_50(HL-60)_/GI_50(*T. brucei)*_; MIC and GI_50_ ratios were calculated from MIC and GI_50_ values shown in Table 1.

**Table 4 molecules-27-05876-t004:** Scoring results of molecular docking of compound **4** to potential enzyme targets and predicted free energies of the binding obtained from the molecular dynamics simulations.

Target	Pose	GOLDScore	CHEMScore	MM-PBSA Binding Energy (kcal/mol) ^1^
G6PD substrate binding site	1	11.65	3.42	4.02
	2	18.08	2.81	−3.98
	3	41.00	2.81	0.49
G6PD cofactor binding site	1	48.14	15.45	−7.25
	2	38.90	11.85	0.71
	3	34.68	10.65	−5.71
PKA1	1	45.45	14.07	−8.09
	2	40.13	8.65	−7.93
	3	38.34	8.64	−4.64
FT	1	31.57	12.23	−2.48
	2	34.38	12.11	−6.61
	3	33.83	11.51	−1.95
IleRL	1	22.53	7.97	−4.58
	2	20.68	6.58	−1.76
	3	21.65	6.56	−0.14
TAO with hydroxide anion	1	32.65	16.33	−12.45
	2	52.42	14.07	−10.24
	3	55.03	13.92	−9.38
TAO without hydroxide anion	1	61.06	21.49	−8.02
	2	55.84	19.80	−2.58
	3	54.15	18.20	−5.76

^1^ Predicted free energy of binding.

**Table 5 molecules-27-05876-t005:** ADMET predictions for compound **4** and the reference drug suramin.

Parameter	Compound 4	Suramin
**Physiochemical properties**		
Molecular weight (g/mol)	212.2	1297.28
Rotatable bonds	4	22
H-bond acceptors	5	23
H-bond donors	3	12
Fraction Csp3	0.3	0.04
TSPA (A^2^)	86.99	534.03
**Lipophilicity (Log *P*_o/w_)**		
iLOGP	1.92	−2.33
XLOBP3	1.8	1.54
MLOGP	0.8	3.51
Consensus	1.38	2.64
**Absorption**		
Water solubility (Log *S*)	−2.32	−7.78
Gastrointestinal absorption (%)	93.13	0
Skin permeability (Log K_P_)	−2.819	−2.735
**Distribution**		
Blood–brain permeability (Log BB)	−1.115	−4.044
CNS permeability (Log PS)	−3.362	−4.943
VD_SS_ (human, Log(L/kg))	0.351	−0.02
**Metabolism**		
CYP1A2 inhibitor	No	No
CYP2C9 inhibitor	No	No
CYP2C19 inhibitor	No	No
CYP2D6 inhibitor	No	No
CYP3A4 inhibitor	No	No
**Excretion**		
Total clearance (Log(mL/min/kg))	0.443	−4.246
Renal OCT2 substrate	No	Yes
**Toxicity**		
AMES toxicity	No	No
Max. tolerated dose (human, Log(mg/kg/day))	−0.27	0.438
hERG I inhibitor	No	No
hERG II inhibitor	No	No
Oral rat acute toxicity (LD_50_, mol/kg)	1.993	2.482
Oral rat chronic toxicity (LOAEL, Log(mg/kg_bw/day))	2.399	7.095
Hepatotoxicity	No	No
Skin Sensitization	No	No

**Table 6 molecules-27-05876-t006:** Effect of gallic acid alkyl esters on the motility of proliferating *T. brucei* bloodstream forms in the presence of glucose or glycerol.

Compound (200 μM)	Motility after 5 min ^1^	
	55 mM Glucose	55 mM Glycerol
–	+ + +	+ +
**1**	+ + +	–
**2**	+ + +	–
**3**	+ + +	–
**4**	+ + +	–
**5**	+ + +	–
**6**	+ + +	–
**7**	+ + +	–
**8**	+ + +	–

^1^ *T. brucei* bloodstream forms were incubated in PBS with 55 mM substrate in the presence or absence of gallic acid alkyl esters as indicated. The motility of the cells was microscopically examined. After 15 min incubation, no change in motility was observed.

## Data Availability

Not applicable.

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
