# Peer review of "Gallic Acid Alkyl Esters: Trypanocidal and Leishmanicidal Activity, and Target Identification via Modeling Studies"

_molecules, 2022, doi:10.3390/molecules27185876_

Round 1

Reviewer 1 Report

Introduction should be better detailed and motivated, including a figure about the proposed chemical series and those of the so far compounds exploited to contrast this infection.

A workflow to sum up the applied strategy should be included.

Results: The rational design of the proposed compounds seems to be lacking. It should be explained and motivated.

Molecular studies should be revised taking into account molecular docking studies of known inhibitors prior to running molecular docking of the novel compounds. The two series of results should be compared. Section 2.3 should be modified, the first paragraph could be moved to the experimental section.

Experimental section: Computational studies herein reported should be improved.

Concerning ligand preparation, it is completely lacking. Please clearly detail the in silico building method, the applied energy minimization strategy and the partial charge calculation.

In silico prediction of admet properties could be included.

Author Response

We thank the reviewer for their time in critically reading and their effort to help improve our manuscript.

Introduction should be better detailed and motivated, including a figure about the proposed chemical series and those of the so far compounds exploited to contrast this infection.

Response: We think that we have described the motivation of the study very clearly in the Introduction. We have explained that there is a need for new drug candidates for the treatment of African trypanosomiasis (see lines 33-44). We have also explained why gallic acid esters are an interesting series of compounds to be investigated as antitrypanosomal agents (see lines 45-56). A figure of the compounds investigated in this study is already included in the manuscript (see Figure 1). What the reviewer means by “and those of the so far compounds exploited to contrast this infection” is unclear to us.

A workflow to sum up the applied strategy should be included.

Response: A workflow of molecular docking explaining the applied strategy is now included at the beginning of section 2.3 (see lines 153-164). It is explained that molecular docking is used only to generate binding hypotheses but not to select the more likely targets of compound 4. The selection of the most likely target is performed from the results of the more accurate MM-PBSA energies.

Results: The rational design of the proposed compounds seems to be lacking. It should be explained and motivated.

Response: The rationale for designing and testing the gallic acid alkyl esters is clearly explained in the Introduction (see lines 45-54).

Molecular studies should be revised taking into account molecular docking studies of known inhibitors prior to running molecular docking of the novel compounds. The two series of results should be compared.

Response: We have now included MM-PBSA calculations with two known inhibitors of TAO. The predicted free energies of binding of these inhibitors are compared to that obtained for compound 4. We performed one additional validation of the docking protocol consisting of the evaluation of its capability of reproducing the experimental TAO-inhibitor complexes (see lines 186-198).

Section 2.3 should be modified, the first paragraph could be moved to the experimental section.

Response: We think that the information provided in the first paragraph of section 2.3 (now it is the second paragraph) includes information important for the understanding of the results. If this information is moved to the Materials and Methods section, it will be difficult for the reader to comprehend the molecular modeling studies. This is also necessary as the Material and Methods section is following the Results section. Therefore, we decided to keep the information provided in the former first paragraph in place. However, we modified the paragraph slightly (see lines 167 and 170).

Experimental section: Computational studies herein reported should be improved.

Response: The experimental section on the computational studies has been updated (see lines 482-485, 492-496, 497-501, 504-510, and 516-532).

Concerning ligand preparation, it is completely lacking. Please clearly detail the in silico building method, the applied energy minimization strategy and the partial charge calculation.

Response: The requested information has been added to the section (see lines 492-496, 497-501, 504-510, and 516-532).

In silico prediction of admet properties could be included.

Response: ADMET predictions are now included in the revised manuscript (see section 2.4, lines 293-313 and Table 5).

Reviewer 2 Report

This study showed the relationship between the chemical moiety of gallic acid and biological activity towards trypanocidal and leishmanicidal using the integration of experimental and computational studies. The authors confirmed the anti-trypanocidal activity of some gallic acid derivatives but not in leishmania. These compounds are safe based on a cytotoxicity study. This manuscript is a clear explanation that easy to understand. However, we have some questions about this work.

For the SAR study, this work focused on only the predicted log P parameter and concluded that the lipophilicity of these compounds is not correlated with the GI50 values. This work should be added more compound descriptors because the other descriptors predicted by some tools might be more correlated with the GI50 values than log P. Moreover, PAIN and drug-likeness of these compounds should be included in this part.

In the section on target identification, the authors mentioned that "gallic acid alkyl esters share some structural similarities with the classical TAO inhibitors salicylhydroxamic acid (SHAM) and ascofuranone" we need statistical data or values to confirm the similarity.

This study tries to identify the human and parasite targets of the potent compounds by using the homology-based target fishing concept [ref. 25]. The protein with higher than 75% sequence similarity will be picked up for the next step. Why don't you focus only on the binding region of protein instead of the non-specify region? 

It might be tough to determine the possible target of a potent ligand by comparing the docking score among the different proteins because amino acids in the binding pocket are not similar. You might consider this point again. 

Any correlation between GOLDScore, CHEMScore, and MM-PBSA?

The entropy can provide ligand stability in the binding pocket. Does MM-PBSA calculation include entropic terms, and how are these values?

Could you provide the figure of the compound 4-binding conformation and 2D interaction of all systems in docked structure and MD trajectory in the supplementary details?

We also want to know the difference between each pose you mentioned in Table 4 because each pose's binding energy and docking score shows a huge difference. 

In terms of ligand-binding stability, only the 4 ns-MD trajectory is insufficient to explain this point. this might be the weak point of this study.

Author Response

This study showed the relationship between the chemical moiety of gallic acid and biological activity towards trypanocidal and leishmanicidal using the integration of experimental and computational studies. The authors confirmed the anti-trypanocidal activity of some gallic acid derivatives but not in leishmania. These compounds are safe based on a cytotoxicity study. This manuscript is a clear explanation that easy to understand. However, we have some questions about this work.

Response: We thank the reviewer for their supportive comments and their suggestions to improve our manuscript.

For the SAR study, this work focused on only the predicted log P parameter and concluded that the lipophilicity of these compounds is not correlated with the GI50 values. This work should be added more compound descriptors because the other descriptors predicted by some tools might be more correlated with the GI50 values than log P.

Response: We have added a correlation analysis between the water solubility (log S) and the GI50 values, which showed a weak association between this descriptor and the trypanocidal activity of the compounds (see lines 111-116 and Figure 2B). Other descriptors (e.g., molar refractivity or topological polar surface area) did not reveal a correlation with GI50 values.

Moreover, PAIN and drug-likeness of these compounds should be included in this part.

Response: Information about PAINS and drug-likeness has been now included in the text (see lines 296-303, Table 5, and lines 314-316).

In the section on target identification, the authors mentioned that "gallic acid alkyl esters share some structural similarities with the classical TAO inhibitors salicylhydroxamic acid (SHAM) and ascofuranone" we need statistical data or values to confirm the similarity.

Response: Tanimoto coefficients have been calculated and mentioned in the text. The coefficient indicates medium similarity between gallic acid alkyl esters and SAHM and the 4-alkoxybenzoic acid ACB41, respectively, and low similarity between gallic acid alkyl esters and ascofuranone (see lines 174-179).

This study tries to identify the human and parasite targets of the potent compounds by using the homology-based target fishing concept [ref. 25]. The protein with higher than 75% sequence similarity will be picked up for the next step. Why don't you focus only on the binding region of protein instead of the non-specify region?

Response: Ideally, the detection of homologous proteins in T. brucei should be only performed with residues in the proteins’ binding sites. However, to the best of our knowledge, no method is available to automatically screen whole proteomes and find homolog proteins based on the identity of the binding sites alone. Such a method would require a database with annotated binding cavities for all proteins for which a sequence exists and the restriction of the Blast search to these regions. This is now mentioned in the manuscript (see lines 482-485).

It might be tough to determine the possible target of a potent ligand by comparing the docking score among the different proteins because amino acids in the binding pocket are not similar. You might consider this point again.

Response: We agree that comparing docking scores among diverse targets can lead to biased results. For this reason, the criterion for selecting the most likely targets of the compound in T. brucei was the free energies obtained with the MM-PBSA method. This is now explained in the manuscript (see lines 153-164 and 207-217).

Any correlation between GOLDScore, CHEMScore, and MM-PBSA?

Response: Kendall’s correlation coefficients between GOLDScore, CHEMScore, and MM-PBSA predictions have been now calculated. The results indicate that there is a correlation between the rankings produced by the three methods. However, the differences in the rankings produced by the three methods suggest that docking scoring functions should not be used as a target selection criterion in replacement of a more accurate methodology such as MM-PBSA. This is now discussed in the manuscript (see lines 232-239).

The entropy can provide ligand stability in the binding pocket. Does MM-PBSA calculation include entropic terms, and how are these values?

Response: The entropic term was excluded from the MM-PBSA calculations due to the high computational cost associated with its estimation. For accurately computing the entropic term by employing normal-mode analysis, it is necessary to use many MD snapshots. We analyzed 23 different complexes with MD simulations. Unfortunately, the computation of the entropic terms for this number of complexes in a reasonable time is beyond our computational capabilities. This is a limitation of our work that is now mentioned in the manuscript (see lines 371-377). Despite this limitation, we consider that our modeling results could be the starting point for the future computer-guided optimization of gallic acid aryl esters as trypanocidal agents using more accurate modeling approaches (see lines 377-379).

Could you provide the figure of the compound 4-binding conformation and 2D interaction of all systems in docked structure and MD trajectory in the supplementary details?

We also want to know the difference between each pose you mentioned in Table 4 because each pose's binding energy and docking score shows a huge difference.

Response: The docking predicted poses as well as the interaction diagrams for the 21 complexes presented in Table 4 are now provided as Supplementary Material (see lines 202-204, 537-539, and Supplementary Materials).

In terms of ligand-binding stability, only the 4 ns-MD trajectory is insufficient to explain this point. this might be the weak point of this study.

Response: MD simulations were performed for obtaining conformational ensembles for MM-PBSA calculations. That is, MD simulations were performed with the objective of studying the energetic stability of the predicted complexes. The optimal length of MD simulations for MM-PBSA calculations as well as the use of single or multiple MD replicas are highly discussed topics in the scientific literature. Different authors agree that short (~ 5 ns) MD simulations are sufficient for MM-PBSA calculations. In our study, we perform MD simulations for 21 different complexes, with five MD replicas of 4 ns per complex. This accounts for total simulation times of 20 ns per complex and 420 ns across all complexes. Based on literature reports, we consider that the selected simulation times are adequate to fulfill the objective of estimating the energetic stability of the studied complexes. This is now mentioned in the revised manuscript (see lines 218-228).